# Rates and reasons for hospital readmission after acute ischemic stroke in a US population-based cohort

**Lily W. Zhou** [1]*, **Maarten G. Lansberg** [2], **Adam de Havenon** [3]

**1** Division of Neurology and Vancouver Stroke Program, University of British Columbia, Vancouver, British Columbia, Canada, **2** Stanford Stroke Center, Stanford University, Palo Alto, California, United States of America, **3** Department of Neurology, Yale University, New Haven, Connecticut, United States of America

* lzhou025@uottawa.ca

**Data Availability Statement:** The data used is publicly available administrative data. It can be purchased for use by researchers after appropriate safety and confidentiality training from the Healthcare Cost and Utilization Project Agency for

## Abstract

Hospital readmissions following stroke are costly and lead to worsened patient outcomes. We examined readmissions rates, diagnoses at readmission, and risk factors associated with readmission following acute ischemic stroke (AIS) in a large United States (US) administrative database. Using the 2019 Nationwide Readmissions Database, we identified adults discharged with AIS (ICD-10-CM I63*) as the principal diagnosis. Survival analysis with Weibull accelerated failure time regression was used to examine variables associated with hospital readmission. In 2019, 273,811 of 285,451 AIS patients survived their initial hospitalization. Of these, 60,831 (22.2%) were readmitted within 2019. Based on Kaplan Meyer analysis, readmission rates were 9.7% within 30 days and 30.5% at 1 year following initial discharge. The most common causes of readmissions were stroke and post stroke sequalae (25.4% of 30-day readmissions, 15.0% of readmissions between 30–364 days), followed by sepsis (10.3% of 30-day readmissions, 9.4% of readmissions between 30–364 days), and acute renal failure (3.2% of 30-day readmissions, 3.0% of readmissions between 30–364 days). After adjusting for multiple patient and hospital-level characteristics, patients at increased risk of readmission were older (71.6 vs. 69.8 years, p<0.001) and had longer initial lengths of stay (7.6 vs. 6.2 day, p<0.001). They more often had modifiable comorbidities, including vascular risk factors (hypertension, diabetes, atrial fibrillation), depression, epilepsy, and drug abuse. Social determinants associated with increased readmission included living in an urban (vs. rural) setting, living in zip-codes with the lowest median income, and having Medicare insurance. All factors were significant at p<0.001. Unplanned hospital readmissions following AIS were high, with the most common reasons for readmission being recurrent stroke and post stroke sequalae, followed by sepsis and acute renal failure. These findings suggest that efforts to reduce readmissions should focus on optimizing secondary stroke and infection prevention, particularly among older socially disadvantaged patients.

Healthcare Research and Quality. If a journal or publication is interested in access to data or analytic files, that information including data elements and data structure can be found here: https://hcup-us.ahrq.gov/nrdoverview.jsp.

**Funding:** LWZ received salary support from a project grant from the Canadian Institute of Health Research (RN387091 - 420683). The funders had no role in study design, data collection and analysis, decision to publish, or preparation of the manuscript. ML has no relevant funding to disclose. AdH is funded by NIH-NINDS K23NS105924. The funders had no role in study design, data collection and analysis, decision to publish, or preparation of the manuscript.

**Competing interests:** Dr. de Havenon has received investigator initiated clinical research funding from Regeneron, AMGEN, and AMAG pharmaceuticals, has received consultant fees from Integra and Novo Nordisk, has equity in TitinKM and Certus, and receives author fees from UpToDate. This does not alter our adherence to PLOS ONE policies on sharing data and materials.

## Introduction

Acute ischemic stroke (AIS) is the most common stroke subtype and accounts for 87% of all strokes in the United States (US) [1]. Hospital readmission following AIS has previously been estimated to occur in 17.4% of patients at 30 days and 42.5% of patients at 1 year [2]. The total direct medical cost of stroke is among the top 10 contributors to Medicare expenditure [3], and is projected to more than double from $71.55 billion to $184.13 billion between the years of 2012–2035 [4]. Furthermore, each additional day spend in health care facilities (vs. home) in the 3 months following stroke has been associated with decreased quality of life in addition to increased healthcare costs [5]. Measuring and reducing readmissions is a vital component in improving quality of life after stroke and in managing the staggering growth in stroke-related healthcare expenditure.

To better understand the frequency and the causes of readmission after AIS, we report the rates of readmission by patient-level medical comorbidities, stroke severity, discharge status and socioeconomic status from the 2019 Nationwide Readmissions Database (NRD), which contains 18 million discharges representing 60.4% of all U.S. hospitalizations from 30 states and from all types of insurance payers [6]. The NRD does not allow data linkage across calendar years for the same patient resulting in variable length of follow up for patients admitted in different times of the year. Because of this, previous work using the NRD studied 30-day readmissions following stroke [7, 8]. We describe a novel technique using time-to-event (ie the product limit method) which 1) provides readmission rates using Kaplan-Meyer estimates including survival time contributions from right-censored patients at 30-days and 1-year following hospital discharge, 2) allows data from all patients in the sample to contribute to the estimates (including patients with less than 30-days follow-up), and 3) allows examination of risk factors for readmission using survival analysis."

## Methods

### Cohort, outcomes, and co-variates

We included individuals with AIS (ICD-10-CM I63*) as the principal discharge diagnosis of their baseline hospital admission and excluded patients under age 18 and elective hospital admissions. The primary outcome was any non-elective hospital readmission during 2019. The secondary outcome was readmission with 1) recurrent AIS (ICD 10 codes I63*) as the principal readmission diagnosis. The tertiary outcome was readmission due to Major Adverse Cardiovascular Events (MACE) defined as: 1) AIS, 2) transient ischemic attack (TIA) (G45*, G46*), 3) Peripheral Vascular Disease (I7*), 4) myocardial infarction (MI) (I21*, I22*) or 5) cardiac arrest (I46*) as the principal readmission diagnosis or 6) a hospital readmission ending in death with a principal readmission diagnosis starting with I (diseases of the circulatory system) to capture cardiovascular death.

The study exposures of individual-level medical comorbidities were pre-defined by NRD, which uses the Elixhauser Comorbidity Measure [9], with the exception of the following comorbidities (defined by ICD-10-CM coding): atrial fibrillation (AF) (I48*), smoking/nicotine dependence (I17*), hyperlipidemia (E78.1-E78.5), and epilepsy (G40*). Because our study used de-identified data, it was exempt from IRB approval. The data is publicly available at https://www.hcup-us.ahrq.gov. We report our findings according to the Strengthening the Reporting of Observational Studies in Epidemiology (STROBE) reporting guidelines.

### Statistical analysis

For the primary analysis, we fit time-to-event models. For individuals with the primary outcome of hospital readmission, the exact number of days between the index admission and the

readmission is captured in the NRD. Time to readmission is calculated using this information and length of stay (LOS) of the index admission. Within the NRD, hospital records from patient transfers are combined to avoid counting hospitalization at the second hospital as a readmission. For those without recurrence, NRD only captures the month of discharge. Because the resulting timescale mix of days and months would prevent an accurate time to event model, we imputed the discharge day of index admission for those without the primary outcome. The imputation selected a random discharge day in the discharge month and creates a timescale in days for individuals without a second hospitalization. This novel methodology reduces bias by allowing the right-censoring to assume a standard distribution.

Stratified Cox models were initially used to assess variables associated with readmission but there were significant violations of the proportional hazards assumption. This was confirmed graphically on plots of Schoenfeld residuals and log-log plots of survival (S1 Fig). Because of this, we used Weibull accelerated failure time regression models for survival analysis, which does not assume constant hazard in survival data [10], to examine the association between patient characteristics (demographics, socio-economic variables, comorbidities) and time to hospital readmission.

This was done first in univariate analysis and then after adjusting for patient age, sex, LOS, discharge disposition, hospital bed size using National Inpatient Sample criteria (small/medium/large) [11], location/teaching status (urban teaching, urban non-teaching, rural), and ownership (government, private non-profit, private for-profit), and baseline National Institute of Health Stroke Scale (NIHSS), made available after October of 2016 by the Centers for Medicare & Medicaid Services (using ICD-10-CM code (R29.7*) to be used in conjunction with the diagnosis of AIS (I63*). Admission NIHSS scores are submitted from the admitting hospital's medical record to the National Readmission Database as a component of administrative data [12]. Adjustment for stroke severity using NIHSS submitted in this form to the Healthcare Cost and Utilization Project has previously been shown to have important implications in analysis of the association between poststroke outcome and patient sex or stroke interventions [13].

Baseline NIHSS and LOS were stratified into categories as there was significant non-linearity on the Box-Tidwell test. The Box-Tidwell test was not significant for age, so it was included in the multivariable model as a continuous variable. The study exposures measured at the index admission included hypertension, diabetes, hyperlipidemia, AF, obesity, drug abuse, smoking or nicotine dependence, depression, dementia, epilepsy, malignancy, rural residence, median income by zip code, primary payer.

Principal diagnoses at readmission were divided into categories using the ICD-10-CM code first letter, which corresponds to organ system. Kaplan-Meyer failure curves divided by principal diagnosis at readmissions were calculated and overlaid on failure curves of all cause readmissions. Missing data is noted where presented and trimmed in the analysis. All statistical analyses were conducted in Stata 17.0 (StataCorp, College Station, TX).

## Results

### Rates of hospital readmission

Of the 285,451 initial stroke hospitalizations in the 2019 NRD (median LOS 3 days, IQR 2–7 days; median hospitalization charge $48 504 USD, IQR $27 226–91 089 USD), 4.1% resulted in mortality (Fig 1).

Of the 273,811 patients who survived their initial hospitalization, 22.2% were readmitted to hospital within 2019 (median LOS 4 days, IQR 2–7; median hospitalization charge $40105, IQR $22 115–76 600 USD). Based on Kaplan-Meier analysis, hospital readmission from any cause occurred in 9.7% at 30 days and in 30.5% at 1 year (Fig 2). Readmissions from MACE

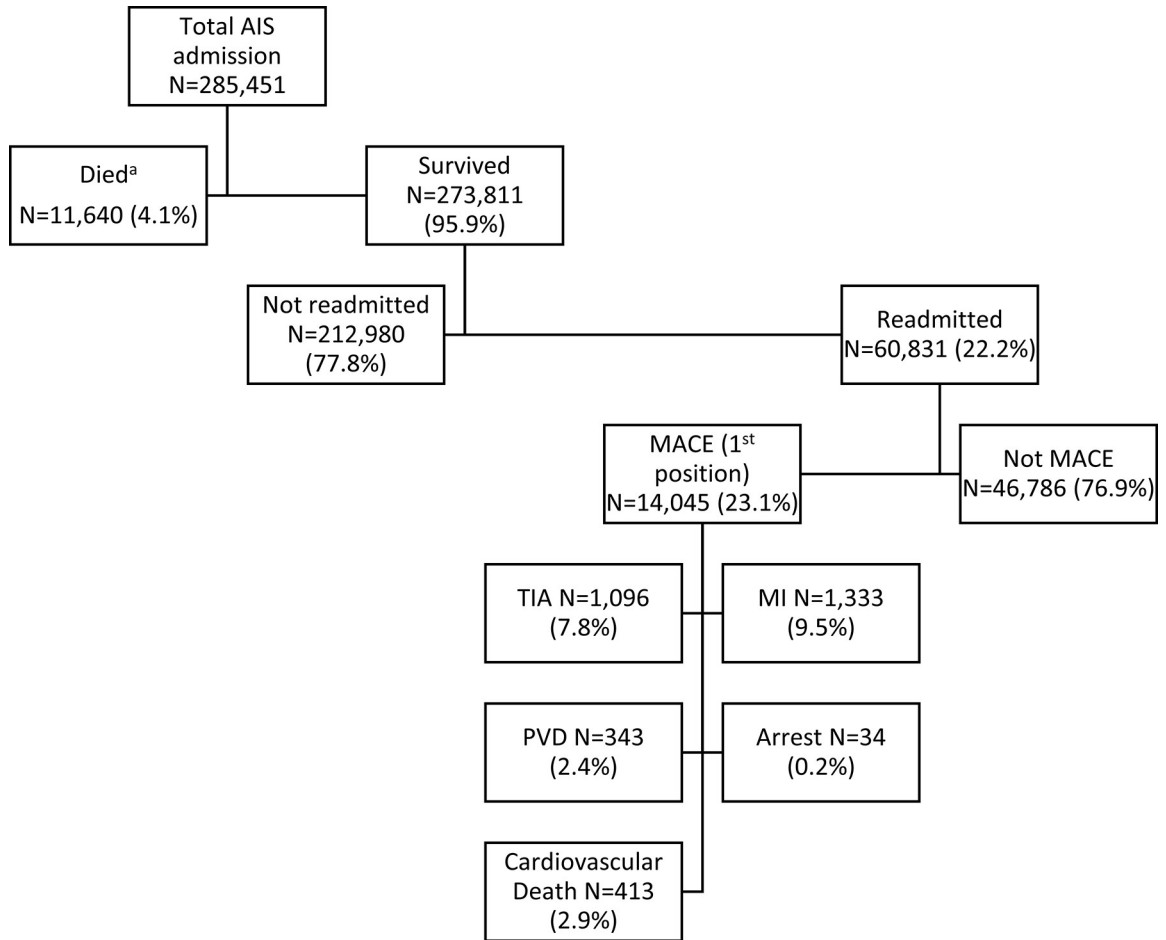

**Fig 1. Outcomes of patients discharged with ischemic stroke and reasons for readmission.** [a] There is no missing data on mortality at discharge within this cohort.

occurred in 2.6% of patients at 30 days and in 7.6% of patients at 1 year after discharge, and due to AIS in 2.2% of patients at 30 days and 5.8% at 1 year.

## Diagnoses at hospital readmission

Twenty three percent of hospital readmissions were due to MACE with AIS accounting for most MACE-related readmissions (77%) (Figs 1, 2 Panel B). Based on the principal ICD 10 code, diseases of the circulatory system were the most common cause for readmission (34.5% %) followed by infectious diseases (10.3%) (Fig 2 Panel A, and S1 Table). Readmission rates by sex, age, stroke severity and length of stay are also shown in Fig 2.

When examining for specific disease diagnoses, the top ten most common diagnosis accounted for 50.2% of all readmissions at 30 days and 44.0% of all readmissions between 30–364 days (Table 1).

The most common causes of readmission were stroke and post stroke sequelae (25.4% of 30-day readmissions, 15% of readmissions between 30–364 days), followed by sepsis (10.3% of 30-day readmissions, 9.4% of readmissions between 30–364 days), and acute renal failure (3.2% of 30-day readmissions, 3.0% of readmissions between 30–364 days).

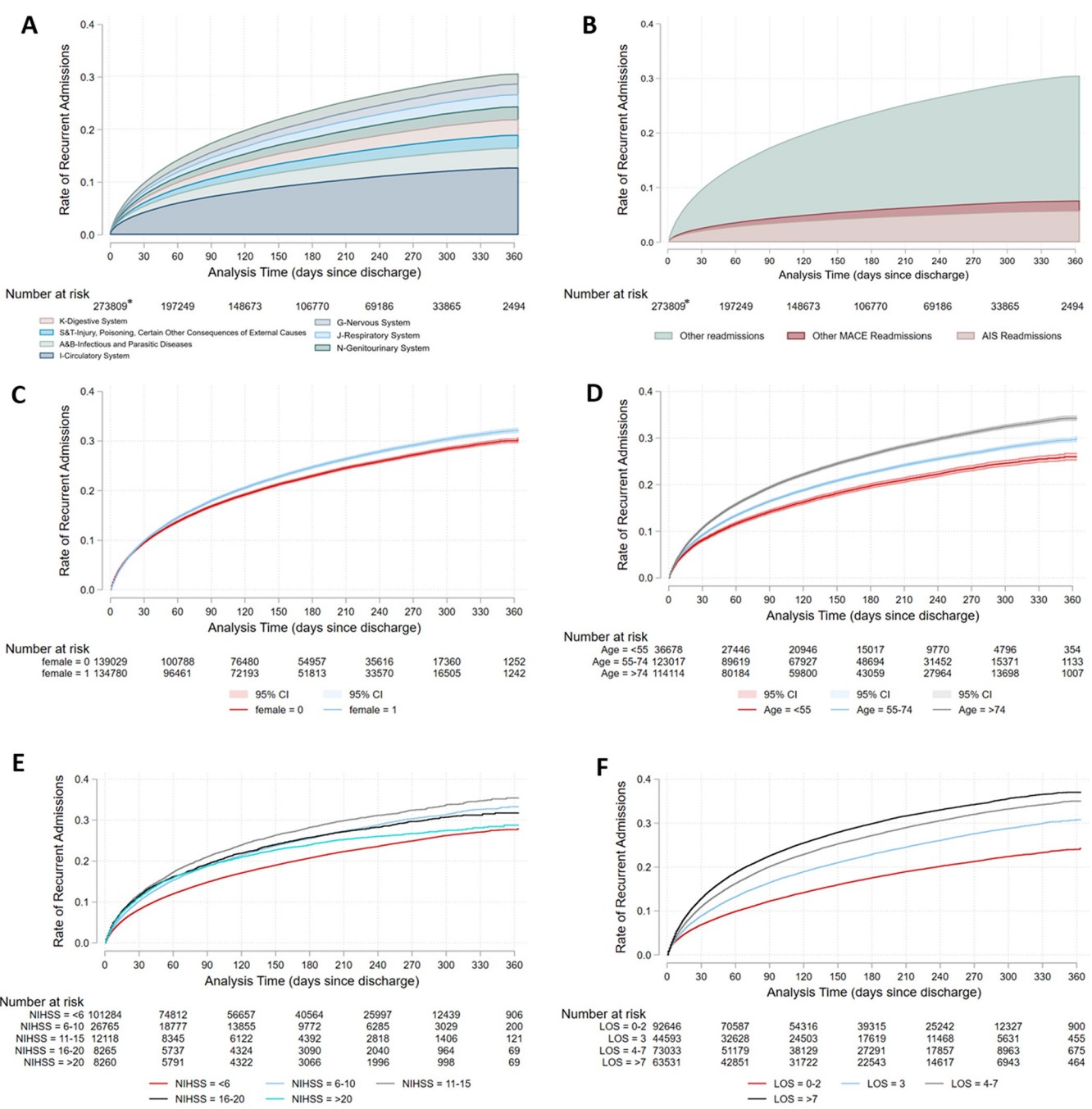

**Fig 2. Kaplan-Meier failure functions for hospital readmission after AIS.** * 2 readmitted patients excluded due to missing length of stay data.

Of the 10,826 individuals with a recurrent stroke admission, 7,667 (70.8%) had an NIHSS documented on readmission. Amongst those individuals, the initial mean±SD NIHSS from their index admission was 5.9±6.6 and with readmission it was 6.7±7.1 (p<0.001). In-hospital mortality on readmission was 4.6% among all cause readmissions and 3.7% among AIS readmissions.

**Table 1. Top 10 diagnosis codes for readmission within first 30 days of discharge vs. after 30 days.**

| | Readmissions within 30 days (n = 25858) | | | | Readmission between 30–364 days (n = 34971) | | | |
|---|---|---|---|---|---|---|---|---|
| Rank | | ICD 10 code | n | % | | ICD 10 code | n | % |
| 1 | I63 | Cerebral infarction | 5594 | 21.6% | I63 | Cerebral infarction | 5232 | 15.0% |
| 2 | A41 | Other sepsis | 2651 | 10.3% | A41 | Other sepsis | 3297 | 9.4% |
| 3 | I69 | Sequelae of cerebrovascular disease | 963 | 3.7% | N17 | Acute renal failure | 1032 | 3.0% |
| 4 | N17 | Acute renal failure | 839 | 3.2% | I13 | Hypertensive heart and renal disease | 1008 | 2.9% |
| 5 | I13 | Hypertensive heart and renal disease | 581 | 2.2% | I21 | Acute myocardial infarction | 892 | 2.6% |
| 6 | J69 | Pneumonitis due to solids and liquids | 498 | 1.9% | N39[a] | Other disorders of the urinary system | 865 | 2.5% |
| 7 | I48 | Atrial fibrillation and flutter | 477 | 1.8% | I11 | Hypertensive heart disease | 840 | 2.4% |
| 8 | N39* | Other disorders of the urinary system | 472 | 1.8% | E11 | Type 2 diabetes mellitus | 799 | 2.3% |
| 9 | I61 | Intracranial hemorrhage | 466 | 1.8% | S72 | Fracture of femur | 737 | 2.1% |
| 10 | I21 | Acute myocardial infarction | 437 | 1.7% | I48 | Atrial fibrillation and flutter | 678 | 1.9% |
| | | Total | | 50.2% | | | Total | 44.0% |

[a]N39.0: Urinary tract infection, site not specified

## Patient and hospital factors associated with hospital readmission

Comparisons between patients with and without readmissions are listed in Table 2.

Patients with readmissions from any cause were more likely to be female, older, and had longer initial LOS. Individuals who were readmitted were also more likely to have traditional vascular risk factors (hypertension, diabetes, and AF) as well as other comorbidities like depression, dementia, epilepsy and a history of drug abuse and malignancy (Table 2). Readmission was more common amongst individuals in the lowest quartile of income, those with Medicaid insurance, those discharged from urban hospitals and those who had a discharge disposition other than home at the end of their index admission (e.g. left against medical advice, discharged to a skill nursing facility). All-cause readmissions were less common among those with a history of smoking or nicotine dependency. Variables associated with readmissions due to AIS or MACE were similar to those associated with all-cause readmissions, with the exception of age, stroke severity, history of atrial fibrillation and dementia, and initial length of stay (Table 2).

Variables associated with time to readmission from all causes are listed in Table 3.

Stroke severity had a non-monotonic effect on risk of all-cause readmission (Table 3 and Fig 2, panel E). Time to readmission was longest in those with NIHSS <6 and shortest in those with NIHSS 11–15. The comorbidity with the strongest association with shortened time to readmission was malignancy, where time to readmission was 0.46 (95% CI 0.44–0.49) times as long as patients without malignancy. Other co-morbidities associated with markedly shorter times to readmission include hypertension (0.68, 95%CI 0.65–0.71), diabetes (0.59, 95%CI 0.57–0.60), AF (0.63, 95%CI 0.61–0.64), drug abuse (0.72, 95%CI 0.67–0.77), depression (0.70, 95%CI 0.67–0.73), dementia (0.67, 95%CI 0.65–0.70), and epilepsy (0.54, 95%CI 0.51–0.57).

Social determinants associated with shorter time to readmission included having Medicare (0.40, 95%CI 0.38–0.41) or Medicaid (0.42, 95%CI 0.40–0.45) insurance compared to private insurance (reference), living in urban settings (0.764, 95%CI 0.74–0.80) and for those living in zip codes with the lowest (0.84, 95% CI 0.82–0.88) and second lowest quartile (0.95, 95% CI0.91–0.99) of median income.

A history of smoking or nicotine dependence was associated with increased time to readmission on univariate analysis but not in multivariable analysis (Model 2). All other associations with medical co-morbidities remained significant in multivariable analysis.

**Table 2. Baseline patient and hospital characteristics shown after stratification by recurrent admission from any cause, MACE events and AIS vs. no recurrent admission.**

| Variable | Any Recurrent Admission vs. No Recurrent Admission | Recurrent MACE Admission vs. No Recurrent MACE Admission | Recurrent Stroke Admission vs. No Recurrent Stroke Admission |
|---|---|---|---|
| | N = 60,831 vs. 212,980 | N = 14,045 vs. 259,766 | N = 10,826 vs. 262,985 |
| Time to readmission in days (median, IQR) | 41 (13–101) vs. N/A | 33 (8–90) vs. N/A | 28 (7–82) vs. N/A |
| Died during readmission | 4.6% vs. N/A | 6.9% vs. N/A | 3.7% vs. N/A |
| Mean age (SD), years | 71.6 (13.3) vs. 69.8 (13.9), p <0.001 | 69.3 (13.4) vs. 70.2 (13.8), p <0.001 | 68.9 (13.3) vs. 70.2 (13.8), p <0.001 |
| Female | 50.8% vs. 48.8%, p <0.001 | 48.7% vs. 49.3%, p = 0.016 | 48.7% vs. 49.2%, p = 0.28 |
| Mean baseline NIHSS (SD) (missing n = 117,118) [a] | 6.4 (6.7) vs. 5.8 (6.6), p <0.001 | 5.2 (5.7) vs. 5.9 (6.7), p <0.001 | 5.3 (5.7) vs. 5.9 (6.6), p <0.001 |
| Mean LOS (SD), days (missing n = 8) [a] | 7.6 (10.0) vs. 6.2 (8.9), p <0.001 | 5.8 (7.8) vs. 6.5 (9.3), p <0.001 | 5.7 (7.8) vs. 6.5 (9.3), p <0.001 |
| Insurance Status | p<0.001 | p<0.001 | p<0.001 |
| Medicare | 73.1% vs. 64.3% | 66.7% vs. 66.2% | 65.5% vs. 66.3% |
| Medicaid | 10.0% vs. 9.2% | 11.4% vs. 9.3% | 11.8% vs. 9.3% |
| Private | 12.0% vs. 19.7% | 15.4% vs. 18.1% | 15.9% vs. 18.1% |
| Self-pay/other | 4.8% vs. 6.6% | 6.4% vs. 6.2% | 6.7% vs. 6.2% |
| Missing (n = 382) [a] | 0.1% vs. 0.1% | 0.1% vs. 0.1% | 0.1% vs. 0.1% |
| Median income by zip code (USD) | p <0.001 | p<0.001 | p<0.001 |
| ≤47,999 | 30.2% vs. 28.1% | 30.9% vs. 28.4% | 30.6% vs. 28.5% |
| 48–60,999 | 25.6% vs. 25.8% | 25.6% vs. 25.7% | 25.6% vs. 25.7% |
| 61–81,999 | 23.7% vs. 24.8% | 23.5% vs. 24.6% | 23.8% vs. 24.6% |
| ≥82,000 | 19.2% vs. 20.0% | 18.7% vs. 19.9% | 18.9% vs. 19.9% |
| Missing(n = 3,608) [a] | 1.3% vs. 1.3% | 1.2% vs. 1.3% | 1.1% vs. 1.3% |
| Hypertension | 88.6% vs. 85.8%, p <0.001 | 89.4% vs. 86.3%, p<0.001 | 89.3% vs. 86.3%, p <0.001 |
| Diabetes | 46.0% vs. 37.3%, p <0.001 | 46.6% vs. 38.8%, p<0.001 | 46.6% vs. 38.9%, p <0.001 |
| Hyperlipidemia | 62.6% vs. 62.2% | 65.7% vs. 62.1%, p<0.001 | 65.4% vs. 62.2%, p <0.001 |
| | p = 0.065 | | |
| Atrial Fibrillation | 29.4% vs. 23.4%, p<0.001 | 22.5% vs. 24.8%, p<0.001 | 21.4% vs. 24.9, p<0.001 |
| Obesity | 15.1% vs. 15.3% | 15.0% vs. 15.3%, p = 0.47 | 15.3% vs. 15.2%, p = 0.85 |
| | p = 0.36 | | |
| Drug Abuse | 3.7% vs. 3.0%, p <0.001 | 3.6% vs. 3.1%, p = 0.002 | 4.0% vs. 3.1%, p <0.001 |
| Smoking or nicotine dependence | 18.5% vs. 19.8%, p <0.001 | 22.0% vs. 19.4%, p<0.001 | 22.8% vs. 19.4%, p <0.001 |
| Alcohol Abuse | 4.6% vs. 4.8%, p = 0.055 | 4.6% vs. 4.8%, p = 0.26 | 4.9% vs. 4.8%, p = 0.53 |
| Depression | 14.1% vs. 11.4%, p <0.001 | 12.4% vs. 12.0%, p = 0.22 | 12.4% vs. 12.0%, p = 0.28 |
| Dementia | 14.2% vs. 11.2%, p <0.001 | 10.6% vs. 12.0%, p<0.001 | 10.3% vs. 12.0%, p <0.001 |
| Epilepsy | 5.6% vs. 3.7%, p <0.001 | 4.1% vs. 4.1%, p = 0.90 | 4.1% vs. 4.1%, p = 1.00 |
| Malignancy | 6.8% vs. 4.3%, p <0.001 | 5.4% vs. 4.8%, P <0.001 | 5.6% vs. 4.8%, p <0.001 |
| Rural (missing n = 988) [a] | 13.0% vs. 15.2%, p <0.001 | 13.7% vs. 14.8%, p<0.001 | 13.2% vs. 14.8%, p <0.001 |
| Discharge disposition | p <0.001 | p<0.001 | p <0.001 |
| Routine/home | 31.3% vs. 45.2% | 40.6% vs. 42.2% | 41.0% vs. 42.2% |
| Short-term hospital | 1.8% vs. 1.3% | 2.9% vs. 1.3% | 3.4% vs. 1.3% |
| Skilled facility/other | 42.2% vs. 31.7% | 30.9% vs. 34.2% | 30.1% vs. 34.2% |
| Home health care | 23.2% vs. 20.7% | 23.3% vs. 21.1% | 22.9% vs. 21.2% |
| AMA | 1.6% vs. 1.1% | 2.4% vs. 1.1% | 2.6% vs. 1.1% |
| Hospital Type | p <0.001 | p = 0.016 | p = 0.009 |
| Urban non-teaching | 19.1% vs. 18.8% | 19.8% vs. 18.8% | 19.9% vs. 18.8% |
| Urban teaching | 75.2% vs. 75.1% | 74.5% vs. 75.2% | 74.5% vs. 75.2% |

*(Continued)*

**Table 2.** (Continued)

| Variable | Any Recurrent Admission vs. No Recurrent Admission | Recurrent MACE Admission vs. No Recurrent MACE Admission | Recurrent Stroke Admission vs. No Recurrent Stroke Admission |
| --- | --- | --- | --- |
| | N = 60,831 vs. 212,980 | N = 14,045 vs. 259,766 | N = 10,826 vs. 262,985 |
| Rural | 5.6% vs. 6.1% | 5.7% vs. 6.0% | 5.6% vs. 6.0% |
| Hospital Bedsize | p <0.001 | p = 0.037 | p = 0.17 |
| Small | 15.1% vs. 15.8% | 15.9% vs. 15.7% | 16.1% vs. 15.7% |
| Medium | 27.4% vs. 27.0% | 27.9% vs. 27.1% | 27.6% vs. 27.1% |
| Large | 57.5% vs. 57.2% | 56.2% vs. 57.3% | 56.4% vs. 57.3% |
| Hospital Control | p <0.001 | p = 0.79 | p = 0.33 |
| Government | 10.9% vs. 11.4% | 11.2% vs. 11.3% | 11.5% vs. 11.3% |
| Private, non-profit | 75.4% vs. 76.2% | 75.9% vs. 76.0% | 76.2% vs. 76.0% |
| Private, for profit | 13.6% vs. 12.4% | 12.9% vs. 12.7% | 12.3% vs. 12.7% |

[a] Total missing data for 273,811 patients who survived their initial hospitalization. Data are complete for all variables where no missing data are reported

## Discussion

Our analysis of readmissions following ischemic stroke in the US using the 2019 NRD revealed that unplanned hospital readmissions were frequent, with the most common reasons for readmission being recurrent stroke and post stroke sequalae, followed by sepsis and acute renal failure. We found multiple patient-level factors that are associated with increased hospital readmission including vascular risk factors (hypertension, diabetes, AF), drug abuse, epilepsy, depression, dementia and malignancy as well social determinants (insurance status, residing in urban areas and zip codes with low median income).

Hospital readmission within 30 days has been defined by the centers of Medicare and Medicaid services as an indicator of poor inpatient care and has been linked to payment determination via the Hospital Readmissions Reduction Program (HRRP) [14]. Our results show that in 2019, rates of all cause 30-day readmissions for patients with AIS was 9.7%. Previously, authors studying the 2010–2015 NRD [7] reported an all-cause 30-day readmission rate of 12% with a decreasing trend during the study period consistent with our lower point estimate. Other US estimates vary widely between 6.4% [15] from a large single center cohort up to 21% using 2008 national Medicare claims data [16]. Differences in rates of readmissions by insurance status (with higher readmissions among those insured with Medicare) and by hospital characteristics, as shown in our analyses, likely explain some of the discrepancies between studies.

Causes for hospital readmissions are complex and may be due to failure to plan for post stroke needs and to implement appropriate stroke prevention, the development of a post-stroke sequelae, or new issues separate from the patient's initial stroke. In our study, more than 50% of all 30-day readmissions could be attributed to ten causes. Ischemic stroke or post stroke sequelae accounted for 25% of readmissions with a further 1.8% of readmissions from hemorrhagic stroke, a proportion of which may represent hemorrhagic transformation. The next most common cause of readmission was from infections, which accounted for 10.3% of readmissions at 30 days, followed by acute renal failure, MI and complications of vascular risk factors such as hypertension, diabetes, and AF. In a 2016 meta-analysis of 7 US, 2 Chinese and 1 Norwegian studies of hospital readmission within 30 days after stroke, the most common causes for readmission were infection (19.9%), coronary artery disease (17.8%) and recurrent stroke (16.0%) respectively [2]. Our findings and other prior research suggest that the highest yield interventions in transitional care after stroke hospitalization would likely include

**Table 3. Association between variables and time to recurrent admission after AIS hospitalization (using a Weibull accelerated failure time model) in univariate analysis (Model 1) and after controlling for patient age, sex, NIHSS category, LOS, discharge disposition, hospital bed size, location/teaching status, and ownership (Model 2).**

| Variable | Model 1 Coefficient exponentiated | 95% CI Lower | 95% CI Upper | p-value | Model 2 adjusted coefficient exponentiated | 95% CI Lower | 95% CI Upper | p-value |
|---|---|---|---|---|---|---|---|---|
| Age | 0.986 | 0.985 | 0.987 | <0.001 | 0.990 [a] | 0.989 | 0.992 | <0.001 |
| Female | 0.883 | 0.861 | 0.907 | <0.001 | 0.965 [a] | 0.931 | 1.001 | 0.055 |
| LOS quartile | | | | | | | | |
| 0–2 | Ref | - | - | - | Ref | - | - | - |
| 3 | 0.614 | 0.588 | 0.640 | <0.001 | 0.727 [a] | 0.686 | 0.770 | <0.001 |
| 4–7 | 0.447 | 0.431 | 0.463 | <0.001 | 0.607 [a] | 0.577 | 0.640 | <0.001 |
| >7 | 0.371 | 0.358 | 0.384 | <0.001 | 0.516 [a] | 0.489 | 0.545 | <0.001 |
| NIHSS | | | | | | | | |
| <6 | Ref | - | - | - | Ref | - | - | - |
| 6–10 | 0.675 | 0.644 | 0.708 | <0.001 | 0.920 [a] | 0.877 | 0.966 | 0.001 |
| 11–15 | 0.567 | 0.532 | 0.603 | <0.001 | 0.868 [a] | 0.814 | 0.925 | <0.001 |
| 16–20 | 0.670 | 0.621 | 0.724 | <0.001 | 1.105 [a] | 1.022 | 1.196 | 0.013 |
| >20 | 0.765 | 0.707 | 0.828 | <0.001 | 1.315 [a] | 1.212 | 1.427 | <0.001 |
| Hospital Type | | | | | | | | |
| Urban non-teaching | Ref | - | - | - | Ref | - | - | - |
| Urban teaching | 1.015 | 0.981 | 1.049 | 0.398 | 1.034 [a] | 0.988 | 1.083 | 0.154 |
| Rural | 1.154 | 1.084 | 1.228 | <0.001 | 1.083 [a] | 0.981 | 1.196 | 0.116 |
| Hospital Bedsize | | | | | - | - | - | - |
| Small | Ref | - | - | - | Ref | - | - | - |
| Medium | 0.914 | 0.877 | 0.953 | <0.001 | 0.948 [a] | 0.893 | 1.006 | 0.078 |
| Large | 0.928 | 0.893 | 0.963 | <0.001 | 0.983 [a] | 0.930 | 1.039 | 0.545 |
| Hospital Control | | | | | - | - | - | - |
| Government | Ref | - | - | - | Ref | - | - | - |
| Private, non-profit | 0.952 | 0.913 | 0.993 | 0.024 | 0.990 [a] | 0.933 | 1.050 | 0.728 |
| Private, for profit | 0.810 | 0.768 | 0.854 | <0.001 | 0.827 [a] | 0.769 | 0.888 | <0.001 |
| Discharge disposition | | | | | | | | |
| Routine/home | Ref | - | - | - | Ref | - | - | - |
| Short-term hospital | 0.293 | 0.265 | 0.324 | <0.001 | 0.335 [a] | 0.291 | 0.385 | <0.001 |
| Skilled facility/other | 0.370 | 0.359 | 0.382 | <0.001 | 0.496 [a] | 0.472 | 0.521 | <0.001 |
| Home health care | 0.481 | 0.464 | 0.498 | <0.001 | 0.592 [a] | 0.562 | 0.624 | <0.001 |
| AMA | 0.309 | 0.278 | 0.344 | <0.001 | 0.272 [a] | 0.234 | 0.316 | <0.001 |
| Hypertension | 0.679 | 0.651 | 0.707 | <0.001 | 0.785 | 0.741 | 0.831 | <0.001 |
| Diabetes | 0.588 | 0.573 | 0.604 | <0.001 | 0.623 | 0.601 | 0.646 | <0.001 |
| Hyperlipidemia | 0.967 | 0.941 | 0.993 | 0.014 | 0.960 | 0.925 | 0.997 | 0.033 |
| Atrial fibrillation | 0.627 | 0.609 | 0.645 | <0.001 | 0.758 | 0.727 | 0.790 | <0.001 |
| Obesity | 1.004 | 0.968 | 1.041 | 0.851 | 0.957 | 0.910 | 1.006 | 0.083 |
| Drug abuse | 0.722 | 0.674 | 0.774 | <0.001 | 0.683 | 0.619 | 0.754 | <0.001 |
| Smoking or nicotine dependence | 1.135 | 1.098 | 1.174 | <0.001 | 0.966 | 0.921 | 1.013 | 0.154 |
| Alcohol abuse | 1.073 | 1.009 | 1.142 | 0.026 | 1.125 | 1.032 | 1.225 | 0.007 |
| Depression | 0.699 | 0.674 | 0.726 | <0.001 | 0.761 | 0.723 | 0.801 | <0.001 |
| Dementia | 0.672 | 0.647 | 0.698 | <0.001 | 0.918 | 0.869 | 0.971 | 0.003 |
| Epilepsy | 0.542 | 0.512 | 0.574 | <0.001 | 0.595 | 0.548 | 0.645 | <0.001 |
| Malignancy | 0.464 | 0.441 | 0.489 | <0.001 | 0.525 | 0.487 | 0.565 | <0.001 |
| Urban residence | 0.764 | 0.735 | -0.230 | <0.001 | 0.782 | 0.739 | 0.827 | <0.001 |

(*Continued*)

**Table 3.** (Continued)

| Variable | Model 1 Coefficient exponentiated | 95% CI Lower | 95% CI Upper | p-value | Model 2 adjusted coefficient exponentiated | 95% CI Lower | 95% CI Upper | p-value |
|---|---|---|---|---|---|---|---|---|
| Median income | | | | | | | | |
| ≤47,999 | 0.848 | 0.815 | 0.881 | <0.001 | 0.832 | 0.790 | 0.877 | <0.001 |
| 48–60,999 | 0.947 | 0.911 | 0.986 | 0.007 | 0.919 | 0.870 | 0.969 | 0.002 |
| 61–81,999 | 0.996 | 0.957 | 1.038 | 0.863 | 0.993 | 0.941 | 1.048 | 0.794 |
| ≥82,000 | Ref | - | - | - | Ref | - | - | - |
| Insurance status | | | | | | | | |
| Medicare | 0.395 | 0.379 | 0.412 | <0.001 | 0.470 | 0.442 | 0.502 | <0.001 |
| Medicaid | 0.421 | 0.398 | 0.445 | <0.001 | 0.483 | 0.448 | 0.522 | <0.001 |
| Private | Ref | - | - | - | Ref | - | - | - |
| Self-pay/other | 0.772 | 0.720 | 0.829 | <0.001 | 0.780 | 0.710 | 0.856 | <0.001 |

[a] Coefficients in base multi-variable model with patient age, sex, NIHSS category, LOS, discharge disposition, hospital bed size, location/teaching status, and ownership without other co-variates

improved infection prevention strategies at discharge and careful transition of responsibility for vascular risk factor management from the inpatient to outpatient settings [17].

Causes for more adverse outcomes following stroke in women compared to men have previously been described and include older age and more severe strokes [18–21]. We found that the higher rate of all cause readmission in women was no longer significant after controlling for stroke severity, age, and hospital characteristics, suggesting that a sex-based difference is likely confounded.

All cause readmission as well as readmission for AIS and MACE was higher amongst patients living in zip codes with the lowest quartile of median income and for patients treated in urban hospitals. Higher 30-day readmissions at "safety net hospitals" located in poor and underserved communities has previously been noted when examining multiple HRRP targeted conditions within the US [22]. These findings highlight persistent health disparities in stroke care and aftercare, which are challenging to address on a policy level. Safety-net hospitals treating vulnerable patients have suffered disproportionally higher penalties under HRRP [23]. However, adjusting penalties based on the income of treated patients may unintentionally entrench lower standards for care provided to disadvantaged populations [24].

When examining only readmissions due to AIS or MACE, readmitted patients appear paradoxically healthier (younger age, less severe baseline strokes, shorter initial LOS and lower rates of AF and dementia) compared to those not readmitted. This effect was not seen when looking at all cause readmissions and may be explained by the fact that patients readmitted for AIS have fewer co-morbidities that could take coding preference during the readmission compared to patients readmitted for other causes [25]. These findings should not be taken to suggest higher stroke recurrence in younger patients or those without AF but rather that younger and healthier patients have less complicated readmissions where stroke recurrence or stroke complications are more likely to be the principal diagnosis.

Our study has limitations. Firstly, the NRD does not allow for linkage of admissions across multiple years or admissions of the same patient occurring across multiple states. We used a novel approach to modelling recurrent ischemic stroke admission in the US using time-to-event analysis, which accurately models the time to right-censoring allowing us to make use of all available data unlike previous works using NRD data that focused solely on risk factors related to 30 days readmissions. However, the data structure still limits the duration of follow

up and makes it challenging to study preceding admissions. Our current study examines patients following their first admission with AIS in 2019 but we were unable to differentiate between first ever stroke vs. recurrent stroke admissions in patients with stroke admissions before 2019.

Secondly, for readmissions due to AIS (ICD code I63*), which accounts for more than 20% of readmissions before 30 days, there is no reliable way of differentiating between incident new AIS vs. readmission due to the index event for psycho-social reasons or recrudescence of symptoms. Among AIS readmissions with a documented NIHSS, the NIHSS on readmission is slightly higher compared to the index admission (6.7 vs. 5.9) suggesting worsening of neurological function and the validity of a recurrent stroke diagnosis. Similarly, there has been no validated methods of examining for hemorrhagic transformation during AIS admissions so the effect of this important prognostic factor could not be analyzed.

We found hospital readmission from MACE occurred in 2.6% of patients at 30 days and in 7.6% of patients at 1 year after discharge. Within MACE, readmissions due to AIS occurred in 2.2% and 5.8% of patients at 30 days and 1 year respectively. Rates of recurrent stroke vary significantly based on stroke etiology. Within the POINT study, of patients with minor stroke and high-risk TIA, the recurrence of ischemic stroke, MI, or ischemic vascular death occurred in 4.9% of patients at 30 days [26]. In THALES, the rate of ischemic stroke at 30 days was 5.6%. Annual stroke recurrence rates reported from clinical trials vary from 2.4%, as seen in the SPS3 trial [27] of patients with lacunar infarction, up to 15%, seen in the WASID [28] and SAMMPRIS [29] trials of patients with intracranial atherosclerotic disease. Rates of recurrent stroke within 5 years after first stroke also varies significantly by age, sex and ethnicity ranging from 5% in black men under 65 years old to 22% in black women aged 65–74 [1]. Unfortunately, the NRD does not contain data on race/ethnicity to examine for differences within these subgroups.

Additionally, there is no way of linking anonymized NRD data to other data sources to obtain information on out-of-hospital mortality. In our study, patients with very high NIHSS at their baseline admissions had lower rehospitalization rates which was likely due to increased rates of out of hospital death following discharge. Out of hospital deaths can account for up to 78% of deaths after stroke at 30 days and 49% of deaths at 1 year [30]. Out of hospital death information would have also enabled competing risk analyses [31] and for a more accurate description of cardiovascular death in the MACE composite. Finally, the cohort being studied was identified using ICD diagnosis codes which is subject to human error during the coding process. However, previous work shows that codes for both AIS and AIS risk factors have acceptable accuracy in administrative data for hospitalized patients [32].

## Conclusion

Unplanned readmission to US hospitals following AIS was high in 2019, with 9.7% of patients being readmitted within 30 days of discharge and 30.5% readmitted by 1 year. The most common causes of readmission were recurrent stroke and post stroke sequelae, followed by sepsis and acute renal failure. Our findings also highlight significant continued health disparities by social determinants of health with higher readmissions among those living in zip-codes with the lowest median income, those living in urban settings and those with Medicare insurance.

## Supporting information

**S1 Fig. Example diagnostic plots showing violations of assumption of proportional hazards for hypertension with top panel showing non-parallel lines on log-log plot of survival and bottom panel showing increasing scaled Schoenfeld residuals over time (more**

**prominent in early period).**
(PDF)

**S1 Table. Principle diagnosis at readmission after discharge with ischemic stroke by ICD category.**
(PDF)

## Acknowledgments

We would like to acknowledge all of the Healthcare Cost and Utilization Project Data Partners that contributed to the data made available for analysis. More information can be found at www.hcup-us.ahrq.gov/hcupdatapartners.jsp

## Author Contributions

**Conceptualization:** Lily W. Zhou, Maarten G. Lansberg.

**Data curation:** Lily W. Zhou, Adam de Havenon.

**Formal analysis:** Lily W. Zhou, Adam de Havenon.

**Methodology:** Lily W. Zhou, Maarten G. Lansberg, Adam de Havenon.

**Resources:** Lily W. Zhou, Adam de Havenon.

**Supervision:** Maarten G. Lansberg.

**Visualization:** Lily W. Zhou, Adam de Havenon.

**Writing – original draft:** Lily W. Zhou, Adam de Havenon.

**Writing – review & editing:** Lily W. Zhou, Maarten G. Lansberg, Adam de Havenon.

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
