## [Decision Letter · Decision Letter 0]

19 May 2023

PONE-D-23-02024Rates and Reasons for Hospital Readmission after Acute Ischemic Stroke in a US Population-Based CohortPLOS ONE

Dear Dr. Zhou,

Thank you for submitting your manuscript to PLOS ONE. After careful consideration, we feel that it has merit but does not fully meet PLOS ONE’s publication criteria as it currently stands. Therefore, we invite you to submit a revised version of the manuscript that addresses the points raised during the review process.

We look forward to receiving your revised manuscript.

Kind regards,

Karthik Gangu

Academic Editor

PLOS ONE

3. Thank you for stating the following in the Acknowledgments/ Funding Section of your manuscript:

“LWZ received salary support from a fellowship grant from StrokeNet (NINDS U24NS107220) and a project grant from the Canadian Institute of Health Research (RN387091 - 420683). Dr. de Havenon is funded by NIH-NINDS K23NS105924.”

 “LWZ received salary support from a fellowship grant from StrokeNet (NINDS U24NS107220) and a project grant from the Canadian Institute of Health Research (RN387091 - 420683).  The funders had no role in study design, data collection and analysis, decision to publish, or preparation of the manuscript.

ML has no relevant funding to disclose.

AdH is funded by NIH-NINDS K23NS105924.  The funders had no role in study design, data collection and analysis, decision to publish, or preparation of the manuscript.”

“Dr. de Havenon has received investigator initiated clinical research funding from Regeneron, AMGEN, and AMAG pharmaceuticals, has received consultant fees from Integra and Novo Nordisk, has equity in TitinKM and Certus, and receives author fees from UpToDate.”

6. We note that you have indicated that data from this study are available upon request. PLOS only allows data to be available upon request if there are legal or ethical restrictions on sharing data publicly. For more information on unacceptable data access restrictions, please see http://journals.plos.org/plosone/s/data-availability#loc-unacceptable-data-access-restrictions.

7. Please note that in order to use the direct billing option the corresponding author must be affiliated with the chosen institute. Please either amend your manuscript to change the affiliation or corresponding author, or email us at plosone@plos.org with a request to remove this option.

Reviewers' comments:

Reviewer's Responses to Questions

**Comments to the Author**

1. Is the manuscript technically sound, and do the data support the conclusions?

Reviewer #1: Partly

Reviewer #2: Yes

Reviewer #3: Partly

2. Has the statistical analysis been performed appropriately and rigorously? 

Reviewer #1: Yes

Reviewer #2: I Don't Know

Reviewer #3: Yes

3. Have the authors made all data underlying the findings in their manuscript fully available?

Reviewer #1: Yes

Reviewer #2: Yes

Reviewer #3: No

4. Is the manuscript presented in an intelligible fashion and written in standard English?

Reviewer #1: Yes

Reviewer #2: Yes

Reviewer #3: Yes

5. Review Comments to the Author

Reviewer #1: 1. In methods section, this sentence needs to be corrected: "Secondary outcomes included readmission with 1) recurrent AIS (ICD 10 codes I63*) as the principal discharge diagnosis, and 2) readmissions due

to Major Adverse Cardiovascular Events (MACE) defined as: 1) AIS, 2) transient

ischemic attack (TIA) (G45*, G46*), 3) Peripheral Vascular Disease (I7*), 4) myocardial

infarction (MI) (I21*, I22*) or 5) cardiac arrest (I46*) as the principal discharge diagnosis

or 6) a hospital readmission ending in death with a principal discharge diagnosis starting

with I (diseases of the circulatory system) to capture cardiovascular death". It is not readmission outcome with discharge diagnosis, rather it is readmission outcome with principal readmission diagnosis. Also correct in abstract and in other parts of manuscript.

2. Authors should explain in details how they got NIHSS scale in methodology.

3. In Abstract, please mention the cut off for age (older) and LOS.

4. In Abstract, please mention the comparison groups of social determinants of health.

5. ICD coding errors are major limitations of National databases and it should be mentioned in the limitations section.

6. Authors should mention the missing data in Figure-1.

Reviewer #2: The manuscript is well written and highlights one major reasons for hospital readmission. A few ideas to include in the outcomes includes, the cost of readmission. Previous studies have been done on national readmission database have compared the trends over time from 2010-2025. It would be interesting to see the trends thereafter from 2016-2020, whether there was any impact of COVID-19 infection on readmission post stroke. Lastly, the tables needs to be accurately labelled and missing values needs to be updated. They can cut short the tables to include the variables relevant to the study.

Reviewer #3: Overall, this peer-reviewed article provides valuable insights into the rates, causes, and risk factors associated with hospital readmissions following acute ischemic stroke in the United States. The reviewer has following concerns:

1.Multiple studies have been published on the 30-day readmission rate following acute ischemic stroke using the same database. The authors should address how their manuscript differs from the already published literature. Additionally, they should consider trend analysis in the 30-day readmission rate, as it decreased from 12% in 2013 to 9.7% in 2019.

https://jamanetwork.com/journals/jamanetworkopen/fullarticle/2696869

https://www.ahajournals.org/doi/full/10.1161/STROKEAHA.116.016085

2.Can the authors include the number of patients who had hemorrhagic conversion following AIS during the index stay?

3.The NRD does not track patients beyond the calendar year. For example, patients admitted in December 2019 cannot be tracked for 30-day readmission as there is no link between the year 2019 and 2020. Therefore, December month admissions should be excluded because the 30-day readmission rate cannot be calculated. Many studies have been published using the same methodology. The authors should clarify if they have excluded December month admissions and add it to the limitations.

4.The NRD provides information regarding inter-hospital transfers. If a patient was admitted to a small hospital and transferred to a large hospital, there is a small chance that the patient might be included twice. Therefore, inter-hospital transfers must be excluded. If the authors have excluded them, please clarify in the methods section. If they have not excluded them, please provide a reason for not doing so.

6. PLOS authors have the option to publish the peer review history of their article (what does this mean?). If published, this will include your full peer review and any attached files.

Reviewer #1: No

Reviewer #2: No

Reviewer #3: No

---

## [Author Response · Author response to Decision Letter 0]

5 Jul 2023

The authors of the present study does not have any special access privileges in accessing this publicly available dataset which other interested researchers would not have. Similarly, we do not have additional information on methodology beyond what is publicly available on the HCUP website: https://hcup-us.ahrq.gov/nrdoverview.jsp. Please see below: 

"The Nationwide Readmissions Database (NRD) is part of a family of databases and software tools developed for the Healthcare Cost and Utilization Project (HCUP). The NRD is a unique and powerful database designed to support various types of analyses of national readmissions for all patients, regardless of the expected payer for the hospital stay. This database addresses a large gap in healthcare data - the lack of nationally representative information on hospital readmissions for all ages. Unweighted, the 2020 NRD contains data from approximately 17 million discharges. Weighted, it estimates roughly 32 million discharges. Developed through a Federal-State-Industry partnership sponsored by the Agency for Healthcare Research and Quality (AHRQ), HCUP data inform decision making at the national, State, and community levels."

With regards to data sharing, the data used is publicly available administrative data. It can be purchased for use by researchers after appropriate safety and confidentiality training from the Healthcare Cost and Utilization Project Agency for Healthcare Research and Quality. We are unable to upload the minimal anonymized data set necessary to replicate the study findings. As a condition of access within the data use agreement, authors agreed that “I will not redistribute HCUP data by posting on any website or publishing in any other publicly accessible online repository. If a journal or publication requests access to data or analytic files, I will cite restrictions on data sharing in this Data Use Agreement and direct them to AHRQ HCUP (www.hcup-us.ahrq.gov) for more information on accessing HCUP data.”

Many thanks to the editor and reviewers for comments to strengthen our paper. We’ve responded to the Reviewer’s Comments below and revised the manuscript with changes below. 

Reviewer #1: 1. In methods section, this sentence needs to be corrected: "Secondary outcomes included readmission with 1) recurrent AIS (ICD 10 codes I63*) as the principal discharge diagnosis, and 2) readmissions due to Major Adverse Cardiovascular Events (MACE) defined as: 1) AIS, 2) transient

ischemic attack (TIA) (G45*, G46*), 3) Peripheral Vascular Disease (I7*), 4) myocardial infarction (MI) (I21*, I22*) or 5) cardiac arrest (I46*) as the principal discharge diagnosis or 6) a hospital readmission ending in death with a principal discharge diagnosis starting with I (diseases of the circulatory system) to capture cardiovascular death". It is not readmission outcome with discharge diagnosis, rather it is readmission outcome with principal readmission diagnosis. Also correct in abstract and in other parts of manuscript.

Thank you for this recommendation for clarification. The manuscript now reads: 

The secondary outcome was readmission due to recurrent AIS (ICD 10 codes I63*) as the principal readmission diagnosis. The tertiary outcome was readmission due to Major Adverse Cardiovascular Events (MACE) defined as: 1) AIS, 2) transient ischemic attack (TIA) (G45*, G46*), 3) Peripheral Vascular Disease (I7*), 4) myocardial infarction (MI) (I21*, I22*) or 5) cardiac arrest (I46*) as the principal readmission diagnosis or 6) a hospital readmission ending in death with a principal readmission diagnosis starting with I (diseases of the circulatory system) to capture cardiovascular death. 

Similar terminology does not appear in the abstract or in other locations of the manuscript. 

2. Authors should explain in detail how they got NIHSS scale in methodology.

The NIHSS scores are submitted from admitting hospitals to the National Readmission Database as a component of administrative data and are extracted from patient records. The following has been added to the methods for clarification. 

“…. baseline National Institute of Health Stroke Scale (NIHSS), made available after October of 2016 by the Centers for Medicare & Medicaid Services (using ICD-10-CM code (R29.7*) to be used in conjunction with the diagnosis of AIS (I63*). Admission NIHSS scores are submitted from the admitting hospital’s medical record to the National Readmission Database as a component of administrative data.[12] Adjustment for stroke severity using NIHSS submitted in this form to the Healthcare Cost and Utilization Project has previously been shown to have important implications in analysis of the association between poststroke outcome and patient sex or stroke interventions.[13]"

3. In Abstract, please mention the cut off for age (older) and LOS.

Based on this comment we realized that our writing was unclear and suggested that we used cutoffs, which is not the case. Instead, we meant to report differences in means. We have tried to clarify this in the revised abstract which now reads: “….patients at increased risk of readmission were older (71.6 vs. 69.8 years, p<0.001) and had longer initial lengths of stay (7.6 vs. 6.2 days, p<0.001).”

4. In Abstract, please mention the comparison groups of social determinants of health.

This abstract now reads: “Social determinants associated with increased readmission included living in an urban (vs. rural) setting, living in zip-codes with the lowest median income, and having Medicare insurance.” These are the main three social determinants. Other insurance status groups (Medicaid, private, self-pay or other) are listed in the main text but not included in the abstract due to word count limitations (300 words max).

The abstract has been revised in other locations to make room within the word limitations for these additions. 

5. ICD coding errors are major limitations of National databases, and it should be mentioned in the limitations section.

This has been acknowledged with the following in the discussion: “Finally, the cohort being studied was identified using ICD diagnosis codes which is subject to human error during the coding process. However, previous work shows that codes for both AIS and AIS risk factors have acceptable accuracy in administrative data for hospitalized patients.[32]" 

6. Authors should mention the missing data in Figure-1.

There is no missing data for Figure 1 as all patients had complete information on death at discharge and all other diagnostic information is coded only when present. This has been clarified at the bottom of the figure. 

 “a There is no missing data on mortality at discharge within this cohort”

Reviewer #2: The manuscript is well written and highlights one major reasons for hospital readmission. A few ideas to include in the outcomes includes, the cost of readmission. 

Thank you for taking the time to improve this manuscript. The following has been added to our manuscript: 

Of the 285,451 initial stroke hospitalizations in the 2019 NRD (median LOS 3 days, IQR 3-7 days; median hospitalization charge $48 504 USD, IQR $27 226-91 089 USD), 4.1% resulted in mortality (Fig 1). Of the 273,811 patients who survived their initial hospitalization, 22.2% were readmitted to hospital within 2019 (median LOS 4 days, IQR 2-7; median hospitalization charge $40 105, IQR $22 115- 76 600 USD).

Previous studies have been done on national readmission database have compared the trends over time from 2010-2025. It would be interesting to see the trends thereafter from 2016-2020, whether there was any impact of COVID-19 infection on readmission post stroke.

We agree these are important elements to consider in our analysis. However, we exclusively analyzed the 2019 National Readmission Data (NRD) for this manuscript so are unable to conduct a temporal trend analysis. Additionally, information from COVID cannot be captured with 2019 NRD data (as the code for COVID, U071, was introduced in HCUP data starting in 2020). 

Lastly, the tables needs to be accurately labelled and missing values needs to be updated. They can cut short the tables to include the variables relevant to the study.

Thank you for this suggestion. The table labels have been checked for accuracy and missing data. Within table 2, the number of missing values has been added for all variables where there were missing values. We have provided a statement in the table’s footnote that ‘Data are complete for all variables where no missing data are reported.’

Reviewer #3: Overall, this peer-reviewed article provides valuable insights into the rates, causes, and risk factors associated with hospital readmissions following acute ischemic stroke in the United States. The reviewer has following concerns:

1.Multiple studies have been published on the 30-day readmission rate following acute ischemic stroke using the same database. The authors should address how their manuscript differs from the already published literature. Additionally, they should consider trend analysis in the 30-day readmission rate, as it decreased from 12% in 2013 to 9.7% in 2019.

https://jamanetwork.com/journals/jamanetworkopen/fullarticle/2696869

https://www.ahajournals.org/doi/full/10.1161/STROKEAHA.116.016085

Thank you for these additional resources. They have been added to our introduction as references 7 and 8. The main difference with the prior studies is that we were able to look at 1-year readmission rates. We have further clarified this difference within our introduction: “The NRD does not allow data linkage across calendar years for the same patient resulting in variable length of follow up for patients admitted in different times of the year. Because of this, previous work using the NRD studied 30-day readmissions following stroke. [7-8] We describe a novel technique using time-to-event (ie the product limit method) which 1) provides readmission rates using Kaplan-Meyer estimates including survival time contributions from right-censored patients at 30-days and 1-year following hospital discharge, 2) allows data from all patients in the sample to contribute to the estimates (including patients with less than 30-days follow-up), and 3) allows examination of risk factors for readmission using survival analysis.” 

We agree that temporal trends in readmission would be of interest, but this manuscript analyzed the 2019 National Readmission Database exclusively and does not have sufficient information for temporal trend analysis. 

2.Can the authors include the number of patients who had hemorrhagic conversion following AIS during the index stay?

We agree with the reviewer that hemorrhagic transformation is an important prognostic predictor in stroke and would be of interest with regards to readmission. 

However, there has been no validated methods for capturing hemorrhagic transformation following acute ischemic stroke within administrative data. One approach is the use of ICD 10 codes for intra-cranial hemorrhage codes and DRG codes related to complications of thrombolysis (61-63). From our experience working with NRD data, this approach leads to rates of hemorrhagic transformation which are not clinically intuitive (such as a hemorrhagic transformation rate of ~10% in AIS following thrombolysis). 

This might, in part, be because patients who had both a primary intra-cranial hemorrhage and an ischemic stroke within the same admission would be miscoded as ‘hemorrhagic transformation’ using this strategy. Because of these concerns, information on hemorrhagic transformation was not included. We have included a statement to the discussion to describe this: “Similarly, there has been no validated methods of examining for hemorrhagic transformation during AIS admissions so the effect of this important prognostic factor could not be analyzed.”

3.The NRD does not track patients beyond the calendar year. For example, patients admitted in December 2019 cannot be tracked for 30-day readmission as there is no link between the year 2019 and 2020. Therefore, December month admissions should be excluded because the 30-day readmission rate cannot be calculated. Many studies have been published using the same methodology. The authors should clarify if they have excluded December month admissions and add it to the limitations.

Thank you for this suggestion for clarification. This is an important limitation of using the NRD using a conventional approach, and why we opted for a survival analysis approach which allows for better use of NRD data compared to previous analysis for stroke readmissions. In our analyses, using Kaplan-Meyer estimates (i.e. the product limit method), patients from December contributed to the 30 days (and 1-year) readmission rate as censored data. In our response to your first comment above, we have detailed how we have modified the introduction to better describe the benefits of using a Kaplan Meyer approach. 

4.The NRD provides information regarding inter-hospital transfers. If a patient was admitted to a small hospital and transferred to a large hospital, there is a small chance that the patient might be included twice. Therefore, inter-hospital transfers must be excluded. If the authors have excluded them, please clarify in the methods section. If they have not excluded them, please provide a reason for not doing so.

This is a valid concern addressed in the NRD data structure by HCUP. “Readmission analyses do not usually allow the hospitalization at the receiving hospital to be counted as a readmission. To eliminate this possibility, pairs of records representing a transfer are collapsed into a single "combined" record in the NRD.” Further information is available from the HCUP website: https://hcup-us.ahrq.gov/db/vars/samedayevent/nrdnote.jsp

We have added a brief statement to the methods to describe this: “Within the NRD, hospital records from patient transfers are combined to avoid counting hospitalization at the second hospital as a readmission.”

---

## [Editor Report · Decision Letter 1]

24 Jul 2023

Rates and Reasons for Hospital Readmission after Acute Ischemic Stroke in a US Population-Based Cohort

PONE-D-23-02024R1

Dear Dr. Zhou,

We’re pleased to inform you that your manuscript has been judged scientifically suitable for publication and will be formally accepted for publication once it meets all outstanding technical requirements.

Kind regards,

Karthik Gangu

Academic Editor

PLOS ONE

---

## [Editor Report · Acceptance letter]

27 Jul 2023

PONE-D-23-02024R1 

Rates and reasons for hospital readmission after acute ischemic stroke in a US population-based cohort 

Dear Dr. Zhou:

I'm pleased to inform you that your manuscript has been deemed suitable for publication in PLOS ONE. Congratulations! Your manuscript is now with our production department. 

Kind regards, 

on behalf of

Dr. Karthik Gangu 

Academic Editor

PLOS ONE